

# On solving textual ambiguities and semantic vagueness in MRC based question answering using generative pre-trained transformers

Muzamil Ahmed[1], Hikmat Khan[1], Tassawar Iqbal[1], Fawaz Khaled Alarfaj[2], Abdullah Alomair[2] and Naif Almusallam[2]

[1] Department of Computer Science, COMSATS University Islamabad, Wah Campus, Wah Cantt, Pakistan
[2] Department of Management Information Systems, School of Business, King Faisal University, Hofuf, Saudi Arabia

Corresponding author
Hikmat Khan,
hikmatullah.khan@namal.edu.pk,
hikmat.ullah@ciitwah.edu.pk

## ABSTRACT

Machine reading comprehension (MRC) is one of the most challenging tasks and active fields in natural language processing (NLP). MRC systems aim to enable a machine to understand a given context in natural language and to answer a series of questions about it. With the advent of bi-directional deep learning algorithms and large-scale datasets, MRC achieved improved results. However, these models are still suffering from two research issues: textual ambiguities and semantic vagueness to comprehend the long passages and generate answers for abstractive MRC systems. To address these issues, this paper proposes a novel Extended Generative Pretrained Transformers-based Question Answering (ExtGPT-QA) model to generate precise and relevant answers to questions about a given context. The proposed architecture comprises two modified forms of encoder and decoder as compared to GPT. The encoder uses a positional encoder to assign a unique representation with each word in the sentence for reference to address the textual ambiguities. Subsequently, the decoder module involves a multi-head attention mechanism along with affine and aggregation layers to mitigate semantic vagueness with MRC systems. Additionally, we applied syntax and semantic feature engineering techniques to enhance the effectiveness of the proposed model. To validate the proposed model's effectiveness, a comprehensive empirical analysis is carried out using three benchmark datasets including SQuAD, Wiki-QA, and News-QA. The results of the proposed ExtGPT-QA outperformed state of art MRC techniques and obtained 93.25% and 90.52% F1-score and exact match, respectively. The results confirm the effectiveness of the ExtGPT-QA model to address textual ambiguities and semantic vagueness issues in MRC systems.

# INTRODUCTION

Machine reading comprehension (MRC) systems aim to resolve the question-answering problem by identifying text spans from one or more passages and giving answers related

to a given context (*Bai et al., 2022*). The MRC evaluates the machine's ability to read, interpret and comprehend structured and unstructured natural language text (*Le et al., 2022*). MRC-based Question Answering (MRC-QA) is one of the active research topics in natural language processing (NLP). The MRC task is challenging in NLP due to the complexity and ambiguity of natural language and is a significant benchmark for measuring a machine's comprehension of natural language (*Li et al., 2022*; *Yang, Sun & Kuang, 2022*). A collection of questions regarding the text is presented to the machine to determine its ability to understand natural language text. The machine's responses are then compared for evaluation to match the gold standard (*Liu, Chen & Xu, 2022*). A machine is considered to have understood the context of its response must meet two criteria: (a) it agrees with human responses, (b) it excludes the irrelevant part of the text (*He et al., 2022*). Every instance in MRC datasets consists of three basic components including a given text passage $P$, a set of questions $Q$ relevant to $P$ and a set of answers $A$. The target of MRC systems is to formulate a mapping function $f = (P, q) \rightarrow a$ where $q \in |Q|$ and $a \in |A|$ that retrieve precise and relevant answers to the query according to the given context (*Lin et al., 2019*).

Previous MRC studies have shown that conventional rule-based language models answer queries from a given context using the constructed rules as a mapping function (*Baradaran, Ghiasi & Amirkhani, 2022*). Such models suffer from two problems: firstly, human effort is required to define the patterns and secondly, such models lack generalization capability (*Liao et al., 2022*). Subsequently, the machine learning models include bootstrapping, morphological interpretation, similarity matching, and Markov methods for natural language comprehension (*Yuan et al., 2021*). However, these methods are not as effective as they lack to extract contextual information and are unable to learn long-range dependencies, so their accuracy can be increased to a certain extent (*Guo et al., 2020*). With the advent of deep learning, models like recurrent neural networks (RNN) and long short-term memory (LSTM) have enabled MRC systems to process sequential data such as text (*Reddy et al., 2020*). The limitation of such models is that they can only learn the past dependencies of the current word, thus not understanding additional linguistic information and leading to low accuracy in the case of complex natural language text.

Unlike sequential models, bidirectional models such as bi-directional LSTM capture both past and future dependencies of text (*Lapchaicharoenkit & Vateekul, 2020*). Moreover, it has been revealed that including explicit syntactic and semantic associations in the attention mechanism provides improved linguistically driven word representations that seem to be favorable for the MRC task (*Chen et al., 2021b*). However, the inclusion of large datasets such as CNN/Daily Mail, SQuAD 1.0, SQuAD 2.0, and Wiki-QA paved the way for further application and thus more learning of deep models in the domain of MRC (*Xu, Tohti & Hamdulla, 2022*). In many of these benchmarks, the answers are usually comprised of a signal entity or short text span. As a result, many queries may be answered quickly and easily by matching words to the context instead of comprehension of the text. To mitigate the concern, diverse benchmark corpuses such as MCTest, RACE, and MultiRC are introduced where the responses can be described in any terms and are not bound to the text spans in the document (*Zeng et al., 2020*). Particularly, a number of questions involve

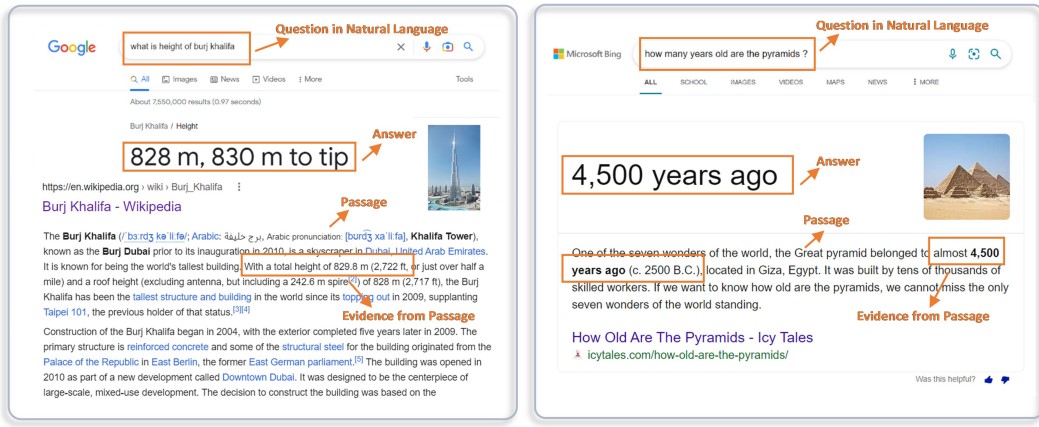

**Figure 1** Real-world applications of machine reading comprehension for search engines.

reasoning which is a sophisticated capacity for comprehension to identify the appropriate response (*Liu et al., 2020*).

MRC systems are divided into two major categories based on generated textual output, including (1) abstractive MRC and (2) extractive MRC systems (*Mohammadi, Ramezani & Baraani, 2022*). Firstly, in the abstracted MRC systems also known as generative mode MRC, the answers are generated following the question and are not always an exact span inside the context. Moreover, these MRC systems are preferably designed for non-factoid questions. Secondly, in extractive MRC also termed selective MRC systems, the answers are extracted exactly from the given passage. These types of systems are preferably suitable for factoid questions (*Sang et al., 2022*). However, generally, a factoid question's answer might be generative, while a non-factoid question's answer might be extractive. For instance, a non-factoid question's answer may consist of a complete sentence that is taken out of its context (*Ji et al., 2022*).

The applications of MRC systems have risen in diverse domains such as search engine optimization, community question answering, customer care centers, conversational agents, health care, education, in the last few years (*Xu, Tohti & Hamdulla, 2022*). In the present era, a vast number of people rely on the internet to extract information related to various fields. Before the advent of MRC systems, the search engines such as Google, Bing, Baidu, Yahoo and Yandex respond to users' queries with a query-specific list of ranked web pages or documents using information retrieval and similarity matching techniques (*Le et al., 2022*; *Li et al., 2022*). The user navigates through retrieved pages and obtains the required information (*Mishra & Jain, 2016*). The MRC systems facilitate the search engines to retrieve precise, concrete, and question-specific answers in natural language as shown in Fig. 1. Moreover, MRC systems give search engines the ability to mimic humans by understanding the context of the posed question and responding to user queries (*Zihayat & Etwaroo, 2021*).

Despite the significant improvements MRC systems over time, due to deep learning and pretrained models they still suffer from linguistic issues such as textual ambiguity and

semantic vagueness. Moreover, understanding the position of the word by considering past and future dependencies in long passages is still a research challenge. Consequently, this research study proposes the transformer model to address linguistic issues with MRC systems. The main objectives and key contributions of this research study are as follows:

- Proposal of novel architecture based on GPT transformers to develop machine reading comprehension-based question-answering system.
- Application of syntax and semantic feature engineering techniques to resolve textual ambiguities and semantic vagueness.
- Learn context-aware representation with past and future text dependencies by involving a positional encoder with an attention mechanism.
- Empirical analysis using three benchmarks MRC datasets to validate the performance of the proposed model and comparative analysis of other methods.

The remainder of this research study is structured as follows: Section 2 presents the relevant literature related to automated question-answering and MRC systems. The proposed research methodology to mitigate the linguistic ambiguities, benchmark datasets, and evaluation measures are presented in Section 3. Next, Section 4 discusses the experimental setup and obtained results of the proposed model. Finally, Section 5 concludes the research work and presents the future work directions.

## RELATED WORK

A number of question-answering systems have been introduced in the last few decades to cope with diverse application domains. *Turing (1950)* proffered a test known as the Turing test to examine whether a machine can think like a human or not. In the Turing test, also called the imitation game, a human can communicate with a machine using query languages to ask a question and the machine responds, and this phenomenon has been discussed from QA perspective in recent studies as well (*Manjunath et al., 2021*). Earlier research on MRC primarily focuses on the identification of text spans for given questions using similarity matching-based techniques. The smart question-answering system used a vectorization technique that applied weighted term frequency-inverse document frequency and cosine similarity to retrieve short and precise answers given in natural language documents (*Soares & Parreiras, 2020*). The authors incorporate a ranking function BM-25 to rank retrieved documents. They achieved an accuracy of 80% on the BNP Paribas corpus containing financial information. However, the dataset used consisted of only 200 queries, which is insufficient to assess the performance and robustness of an MRC model. Moreover, the TF-IDF and cosine similarity do not consider semantic as they focus on frequency-based similarity. The question-answering system lacks the common-sense ability to answer irrelevant and unanswerable questions. To address the reasoning power ability (*Aithal, Rao & Singh, 2021*) compute the similarity between the posed question and possible generated question over a document. The authors used ranked question similarity scores to retrieve answers to irrelevant questions. They applied cosine similarity measures after extracting sentence embedding. They selected 1,000 unanswerable and irrelevant question from the famous SQuAD 2.0 text corpus to evaluate the performance and achieved 85% accuracy.

An automated QA system for waste collection system (*Jiang et al., 2021*) uses CNN deep learning model. The authors applied several natural language text pre-processing techniques such as word segmentation and stop-word removal. They used 128-dimension word2vec to capture feature representation. The CNN architecture consists of four convolution layers with max pooling and soft-max activation functions in the fully connected layer. They prepared their text dataset for the waste collection system and achieved 88.6% accuracy. Another question-answering system to solve problems of financial students combines LSTM and CNN deep learning models (*Chen, Zhong & Zhu, 2022*). The authors also introduce an attention mechanism using Gated Recurrent Unit (GRU) to assign more weights to important words in a sentence. They also capture sentences' past dependencies and semantic features by arranging information in chronological order to comprehend natural language. They analyze the results of a self-prepared dataset for the financial problem with different variations of dropout. The model achieved 82% accuracy with a 0.6 dropout rate.

A non-factoid automated QA system (*Zihayat & Etwaroo, 2021*), used Bi-LSTM and BERT. The Bi-LSTM captures semantic and syntax features from posed questions and answers in the repository. The deep embeddings are captured using BERT end-to-end architecture. The authors used Google patent data for empirical analysis after text pre-processing such as word segmentation, lemmatization, and part-of-speech tagging. The question-answering process involves distance measure and a top-k similar sentence method to retrieve a close and precise answer. They achieved 86.6% accuracy. Multi-Lingual Question Answer (MLQA) is considered tough among other QA types. A MLQA system (*Loginova, Varanasi & Neumann, 2021*) used deep learning architecture to tackle the challenges and improve QA performance. The authors focus on three widely spoken natural languages including Arabic, English, and German. They applied cross-lingual embedding to extract deep features. The question processing according to the target language is hard due to ambiguity and word mismatch challenges. They used shared semantic representation to retrieve cross-lingual questions. They used Insurance QA and Sem Eval datasets and attained MAP scores of 64.48% and 49.39% respectively.

Machine reading comprehension is quite a challenging task under the umbrella of QA. The transformer-based models such as BERT improve the performance of MRC to retrieve precise answers within a given passage. However, BERT does not perform well to retrieve answers from lengthy passages. An MRC-QA (*Galitsky, Ilvovsky & Goncharova, 2021*) involves a self-attention mechanism with pre-trained BERT to enhance the ability of a model to comprehend lengthy passages. The architecture of the model combines the embedding layer with a discourse-aware self-attention layer and dual context aggregation layer. The authors evaluated model performance using four benchmark datasets including SQuAD, News-QA, QuAC, and MSQ, and achieved 90%, 75.05%, 74.88%, and 71.65% accuracy, respectively. Another comprehension model for multi-type questions used a multi-layer transformer-based model (*Chen et al., 2021a*). The authors include an encoding layer for question processing and a decoding layer for generated answer processing. They also fine-tune pre-trained sequences to sequence models with two large corpora including

DROP and QUOREE. These datasets contain about 96,000 question-answer pairs. The semantic aware model attains 81.34% accuracy.

The use of an attention mechanism improves the accuracy of the question-answering system. However, the ensemble transformer-based model (*Matsubara et al., 2022*) explores how to boost the accuracy without increasing model complexity. The authors used multiple head attention mechanism along with shared encoder. Their model comprises two basic units: first contains shared encoding module and ranked multiple heads, and second has transformer-based architecture. They evaluate the model performance on three English language benchmarks dataset including ASNQ, IAS2 and Wiki-QA having about 60,000 questions. The ensemble transformer-based model achieved 70.3%, 75.2% and 89.3% MAP on IAS2, ASNQ and Wiki-QA respectively. Although the attention mechanism significantly improves the QA performance, the semantics and word positions in questions and answers also plays vital role in sentence comprehension. Another, BLSTM, SRLP question-answering system (*Bi et al., 2021*) incorporate the semantic positional attention to assign weights to words in word embedding by considering their position in sentence. The existing approaches extract past dependencies using LSTM deep architecture to comprehend the semantics. The authors used Bi-LSTM deep learning model to capture both past and future dependencies of words in question answering sentence. They achieved about 87% accuracy on food safety text corpus FS-QS.

Inferential MRC aims to answer posed generic question over a passage. Another MRC-QA (*Yu, Zha & Yin, 2019*) involved context-aware embeddings along with attention-based mechanism. In the proposed model authors created an evidence-based chain from given text by using recursive cell-based architecture. They used an inferential network by combining three micro-infer-cells. These micro-infer-cells are linked together and consist of four components including memory, reader, writer, and master. The inference cell rake encoding of question, answer and options as input and generate a precise answer. They used the RACE, MC-Test and Multi-RC corpus for empirical analysis and achieved 86%, 87.7% and 65% accuracy respectively.

The multi-passage MRC retrieves the answer of posed question from multiple passages instead of single passage. Sentence segmentation across multiple documents often responds with poor quality answer. To address this issue MRC-QA (*Lin et al., 2019*) uses paragraph level segmentation approach for answer retrieval in multi-passage MRC. The authors proposed a neural network-based model along with an attention layer for closest paragraph matching. They incorporate Bi-LSTM deep learning architecture to learn past and future dependencies during matching. Moreover, added a POUGE-L as scoring layer in architecture to rank the quality of selected paragraph. They used the Du-Reader version 2 corpus for empirical analysis and achieved 89.4% recall. Predicting answer availability in MRC is essential to access whether posed question is available in passage or not. An inspection model *NeurQuRi* (*Back et al., 2020*) incorporate GRU along with attention-based loss function to predict the answer availability. The framework includes stop-word-removal, Bi-LSTM deep learning model for word embedding, multiple inspector encoder and inspector comparators. The authors used the SQuAD 2.0 famous MRC dataset

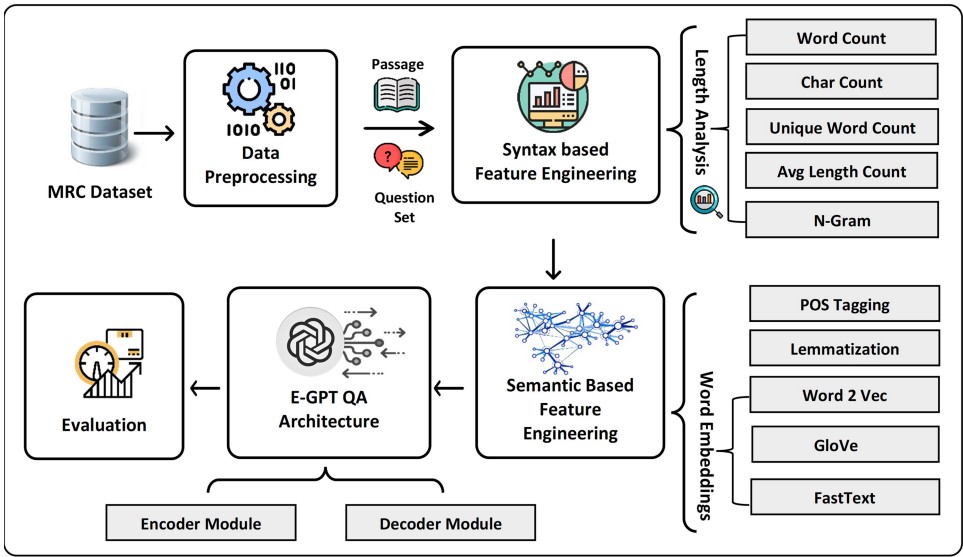

**Figure 2** **The overview of proposed framework for MRC question answering system.**

(*Rajpurkar, Jia & Liang, 2018*) and achieved 86.9%, 86.5% and 89.5% F1-score, accuracy, and EM respectively.

## PROPOSED RESEARCH METHODOLOGY

In this research study, we propose the transformers-based model to mitigate syntactic ambiguities and semantic vagueness with MRC-based question answering. Figure 2 presents the architecture of the proposed model which consists of four main components including data pre-processing, semantic and syntactic feature engineering, and an extended GPT model. In the first step, the extracted text data is preprocessed using NLP techniques and structured as a set of questions and associated with text passages for answering those questions. Next, after tokenization, syntax-based features are extracted based on word frequency and length analysis. Moreover, to attain a better understanding of natural language text, semantic features such as deep learning-based word embedding, lemmatization, and part of speech (POS) tagging are extracted. The core of extended GPT-QA is composed of encoder and decoder components. The novelty of the study includes the proposal of a positional encoder to learn position-aware embeddings for both questions and answer passages. In addition, the decoder module involves both affine and aggregation layers in Generative Pre-trained Transformer (GPT) based architecture to retrieve answers to the posed questions.

### Syntax-based feature engineering

Syntax-based features are significant for natural language text comprehension. In this model, we incorporate length analysis and word frequency features for syntactic analysis. Firstly, we employed a regular expression-based tokenizer (RET) to split text strings into individual tokens. The RET uses regular expressions for context-aware separation of words.

For example, conventional tokenizers split the string *"Let's see how it's working"* into nine tokens such as *[{Let}, {'}, {s}, {see}, {how}, {it}, {'}, {s}, {working}]*. Although RET splits the same string into only five reasonable tokens such as *[{Let's}, {see}, {how}, {it's}, {working}]*. In MRC, length analysis provides more insights with easy to understand computations. In this model, we used four length analysis measures including word count, character count, unique words count, and average word length per question. The sum of tokens is termed as word count ($W^c$) and calculated using Eq. (1):

$$W^c = \sum_{i=0}^{|\text{token}|} i \tag{1}$$

where |token| is a number of tokens and $i$ indicates the individual token. The char count is another length analysis measure that sums the number of characters in each token. The char count ($\text{Char}_{\text{count}}$) is computed using Eq. (2):

$$\text{Char}_{\text{count}} = \sum_{i=0}^{|\text{token}|} \text{len}(\text{token}_i) \tag{2}$$

where the function len() returns the length of individual token $i$. The rare word UWC is a subset of word count where each word is counted as once denoted as {UWc $\subset$ Wc|Wc is counted as once}. The average word length (AWL) is calculated using the following Eq. (3):

$$\text{AWL} = \frac{1}{n} \sum_{i=0}^{|\text{token}|} i \tag{3}$$

where $n$ denotes the number of sentences. Next, we extract word frequency features such as n-grams and bi-grams. The n-grams are a combination of n consecutive words or tokens in a question or answer passage. Similarly, the bigrams are a combination of two consecutive words in a text document. In GPT-QA, these features are significant to find the probability of the occurrence of a specific token after or before a certain word.

## Semantic-based feature engineering

Semantic-base features assist in understanding the semantics and context of natural language text. The interpretation of the posed questions is quite complex due to the involvement of subjectivity and vast complexity in natural language. In this model, we incorporated lemmatization, POS tagging, and word embedding as semantic based features. Lemmatization is the process of transforming words into their meaningful roots also known as a lemma. The lemmatizers understand the actual meaning of words using a knowledge base before the lemma transformation. Afterward, we applied POS tagging for the categorization of individual words according to their part of speech. RNN deep learning model is applied for assigning POS tags. The specific POS tag is also used to predict the probability of a successive word in a sentence as shown in Fig. 3. The list of parts of speech to be considered in the proposed model along with their associated tags and examples is presented in Table 1.

As a next step, word embeddings are learned to transform natural language text into real value vectors. In this phase, each token is mapped to the corresponding vector using a deep

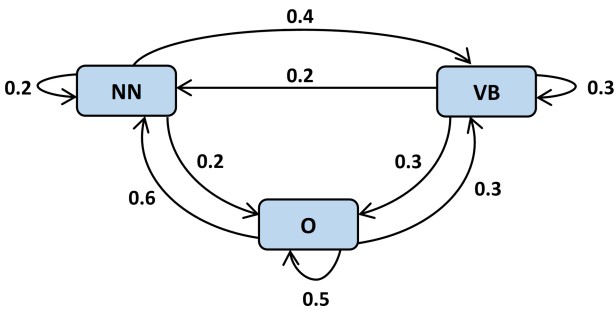

**Figure 3  Utilization of POS tags to predict conditional probability of successive words (*Loginova, Varanasi & Neumann, 2021*).**

**Table 1  List of part of speech to be used in proposed model.**

| Sr. no | Part of speech | Tag | Examples | Sr. no | Part of speech | Tag | Examples |
|---|---|---|---|---|---|---|---|
| 1 | Preposition | PR | Above, ago, aside | 8 | Noun | NN | Child, Asia, bee |
| 2 | Conjunction | CJ | Although, unless, since | 9 | Possessive Pronoun | PPN | Mine, yours, here |
| 3 | Adverb | AD | Towards, abroad, nearby | 10 | Adverbial Participle | ADP | Going |
| 4 | Interjection | IJ | Alas, hey, well | 11 | WH-quest Pronouns | WHQ | Why, what, who |
| 5 | Verb Infinitive | VI | Go on, can't stand | 12 | Verb (Other forms) | VV | Accept, goes, guess |
| 6 | Numerical | NML | Nine, five | 13 | Personal Pronoun | PP | I, we, you, she |
| 7 | Adjective | ADJ | Clean, elegant, big | | | | |

learning model. Three deep word embeddings such as word2vec, GloVe, and fastText are computed as deep features. The word2vec (short form of word to vectors) is an embedding technique for large datasets that is considered a complete model architecture for vector representations. In this work, we use a continuous skip-gram algorithm because it considers the word context to generate vector representation. The conditional probability of context prediction is computed using Eq. (4):

$$P(w_o|w_C) = \frac{\exp\left(u_o^T v_c\right)}{\sum_{i \in v} \exp\left(u_i^T v_c\right)} \tag{4}$$

where $P$ denotes the conditional probability of the word, $o$ and $c$ denote the index on the dictionary and center of the word respectively. The function exp() indicates the exponential of words in the vocabulary. However, the $v$ is a set of vocabulary indexes where $v = \{0, 1, 2, \ldots |v| - 1\}$. The term $u_o$ and $v_c$ denotes words in vocabulary and the $T$ is the length of the question. GloVe, (short form of global vectors) is a widely used deep learning model that involves word co-occurrence for vector representation. Glove embeddings consider the mutual influence of two words on each other by analyzing the frequency of appearance. The vector value of a word using the GloVe model is calculated using Eq. (5):

$$F\left(w_i, w_j, \tilde{w}_k\right) = \frac{P_{(i|k)}}{P_{(j|k)}} \tag{5}$$

where $w_i, w_j$ are words in context and $\tilde{w}_k$ denotes the word that is out of context. The term $P$ denotes the conditional probability of the context $i|k$ and $j|k$ derived from corpus
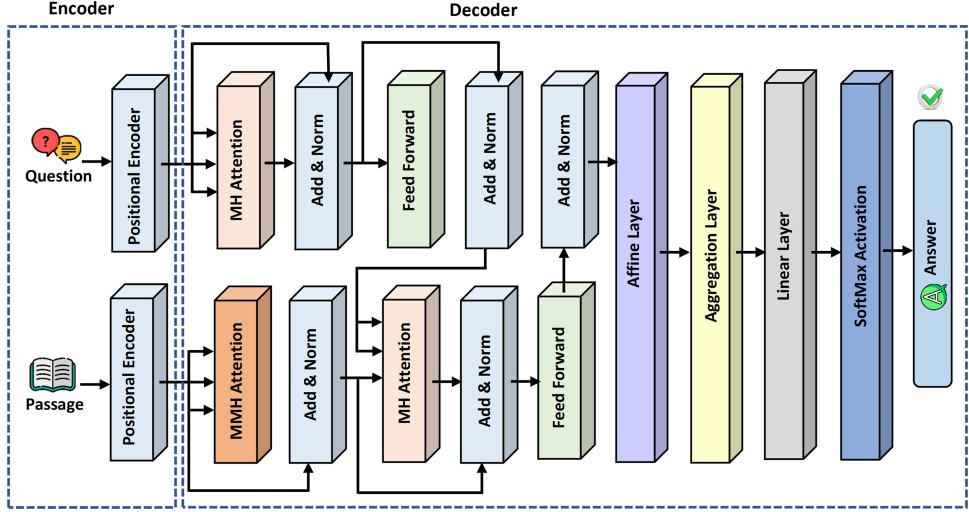

**Figure 4  The architecture of the proposed ExtGPT-QA for machine reading comprehension.**

calculated as Eq. (6):

$$P(i|k) = \frac{X_{ij}}{X_i} \qquad (6)$$

where the $X_{ij}$ frequency of words $i$ and $j$ have occurred together in the corpus. The fastText is the advanced form of word embeddings that uses an n-gram sequence for vector representation instead of individual words.

## Extended generative pre-trained transformer-question answering (ExtGPT-QA)

The novelty of this study is the proposal of the variation of GPT-3 referred to as GPT-QA for retrieval of precise answers to questions from the given passage in the context of machine reading comprehension. The proposed model comprises two basic components: encoder and decoder. In the encoder module, positional encoding is applied on questions and answer passage embeddings separately. Figure 4 presents the detailed architecture of the proposed ExtGPT-QA model for machine reading comprehension.

### Positional encoder

To address the text ambiguities, the positional encoder allocates a unique representation with each word in the sentence for reference. As the baseline model assigns index base numbers which creates problems for longer sentences. The output of the positional encoder is a d-dimensional vector where rows contain positional information of encoded words. Let's suppose, we have an input sentence with $L$ length. The positional encoding of a word on $k$th the position is computed using Eqs. (7) and (8):

| Sentence | Position | $i = 0$ | $i = 0$ | $i = 1$ | $i = 1$ |
|---|---|---|---|---|---|
| I | 0 | $P_{00} = \sin(0)$ $= 0$ | $P_{00} = \cos(0)$ $= 1$ | $P_{00} = \sin(0)$ $= 0$ | $P_{00} = \cos(0)$ $= 1$ |
| am | 1 | $P_{10} = \sin(1/1)$ $= 0.84$ | $P_{11} = \cos(1/1)$ $= 0.54$ | $P_{12} = \sin(1/10)$ $= 0.10$ | $P_{13} = \cos(1/10))$ $= 1.0$ |
| a | 2 | $P_{20} = \sin(2/1)$ $= 0.91$ | $P_{21} = \cos(2/1)$ $= -0.42$ | $P_{22} = \sin(2/10)$ $= 0.20$ | $P_{23} = \cos(2/10)$ $= 0.98$ |
| bot | 3 | $P_{30} = \sin(3/1)$ $= 0.14$ | $P_{31} = \cos(3/1)$ $= -0.99$ | $P_{32} = \sin(3/10)$ $= 0.30$ | $P_{33} = \cos(3/10)$ $= 0.96$ |

**Figure 5** Positional encoding matrix for the sentence "I am a bot" with $d = 4$ and $n = 100$.

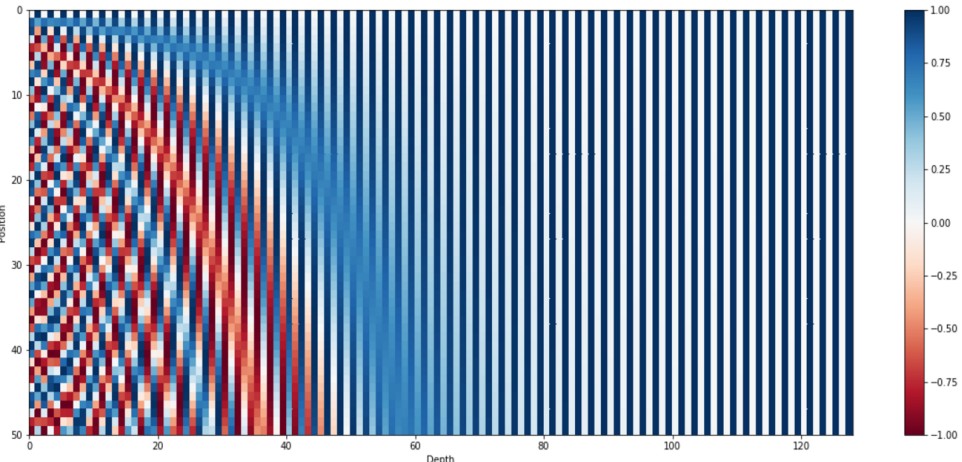

**Figure 6** Positional encoding matrix for sentence length 50 along with $d = 128$ and $n = 10,000$.

$$P(k, 2i) = \sin\left(\frac{k}{n^{\frac{2i}{d}}}\right) \tag{7}$$

$$P(k, 2i+1) = \cos\left(\frac{k}{n^{\frac{2i}{d}}}\right) \tag{8}$$

where $P(k, j)$ is mapping function for words at $k$th position into positional matric. The terms $i$, $n$ and $d$ denote mapping index, user-defined constant, and embedding space dimension respectively. Figure 5 presents positional encoder matric for the sentence "*I am a bot*" with dimensions $d = 4$ and $n = 100$. However, the standard value of n is commonly set up with 10,000 for longer paragraphs with high dimensions. Figure 6 shows the 128-dimension output matrix of the positional encoding layer with $n = 10,000$ for a question passage having a length of fifty words.

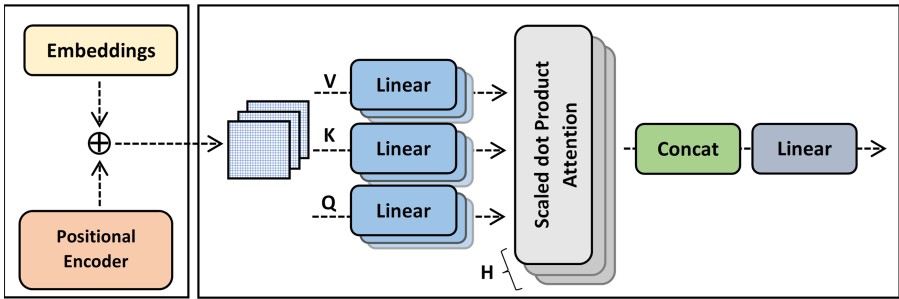

**Figure 7** The architecture of multi-head attention.

## Multi-head (MH) attention layers

The first layer of the ExtGPT-QA decoding module is the multi-head attention layer. The basic purpose of the attention layer is to highlight the valuable parts of a text through by adding weights by extracting the relationship of words. The baseline model adopts a self-attention layer that uses vector representation for average weight computation by proportional weight to similarity score. However, self-attention increases training overhead due to adding extra weight for lengthy passages.

The multi-head attention is a mechanism that concatenates the output of multiple scaled dot product attention runs simultaneously according to the expected dimension. The input of multi-head attention comprises a query $q \in R^{d_q}$, key $k \in R^{d_k}$, and value. $v \in R^{d_v}$. The linear output $h_i (0 \leq i \leq H)$ after concatenation is computed using Eqs. (9) and (10):

$$h_i = f\left(W_i^q q, W_i^k k, W_i^v v\right) \in R^{p_v} \tag{9}$$

$$W_o \begin{bmatrix} h_i \\ \vdots \\ h_H \end{bmatrix} \in R^{p_0} \tag{10}$$

where $f$ is a function to compute scaled dot product attention and $W_i^j J$ denote learnable parameters for query, key, and value respectively. The architecture of multi-head attention is shown in Fig. 7.

## Feed forward layer

In the proposed ExtGPT-QA model, a feed-forward layer is applied after multi-head attention contains the actual weights of the model. The feed-forward layer acts as a function that accounts for position and processes each input matrix separately. This layer has a significant role in language modeling and comprises 2/3 the parameters of the entire model. Let's suppose, we have a vector $x \in R^d$ learned from an input string. The output of the feed-forward layer can be expressed using Eq. (11):

$$FF(x) = f\left(x \times K^T\right) \times V \tag{11}$$

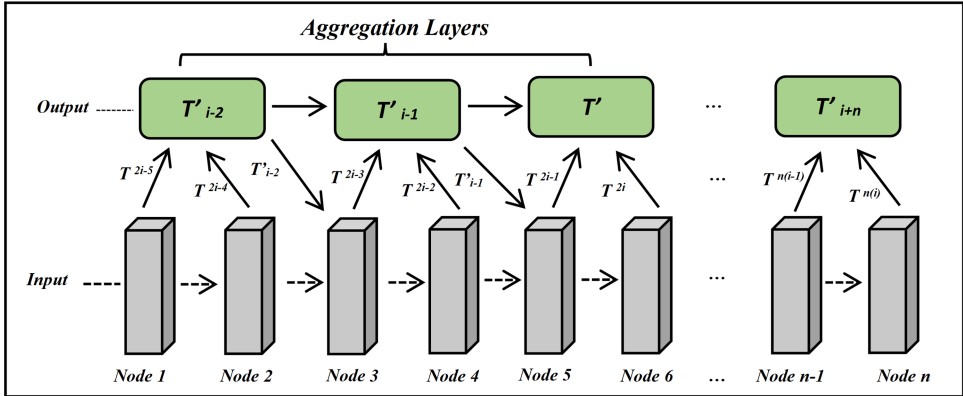

**Figure 8** The architecture of aggregation layer for ExtGPT-QA.

where $f$ denotes the activation function identical to rectified linear activation unit for non-linearity. Moreover, the terms $K$ and $V \in R^{d_m \times d}$ are parameter vectors of the language model and $d_m$ represents couples of key values containing neural network memory.

## Affine and aggregation layers

We introduce the two novel layers: the affine layer and the aggregation layer in the architecture of the GPT model. An affine layer is used to perform a linear transformation on the input vector as a distributed fully connected layer. The distribution of a fully connected layer overcomes the model complexity over huge parameters of a language model. The affine layer links all nodes of the contained layer with subsequent layers and adds bias with each connection. The activation function of an affine layer is computed using Eq. (12):

$$y = f(W \times x + b) \tag{12}$$

where $f$ is an activation function applied over $x$ represents the output of the feedforward layer and two learnable parameters $W$ and $b$ are denoted as associated weights and bias as scaling matrix respectively.

The aggregation layer is introduced into GPT-QA architecture for the generation of semantic-aware answers. The aggregation layer aggregates the nodes' information from the form base layer to the top layer as shown in Fig. 8. It considers semantic and contextual information using a positional matrix learned from the positional encoder. The aggregation of a dimensional matrix is calculated using Eq. (13).

$$\hat{T}^i = \begin{cases} \text{Aggregatoion}\left(T^{2i-1}, T^{2i}\right) & i = 1 \\ \text{Aggregation}\left(T^{2i-1}, T^{2i}, \hat{T}^{i-1}\right) & i > 1 \end{cases} \tag{13}$$

## Datasets description

We evaluated the proposed ExtGPT-QA architecture on three machine reading comprehension datasets: Wiki-QA, SQuAD, and News QA. Wiki-QA short for the Wikipedia question answering dataset was prepared by *Yang, Yih & Meek (2015)* and

comprised 3,047 questions and 29,258 candidate sentences. In addition, a subset of about 1,473 sentences is annotated using mechanical truck workers (MTW) to exactly match answers with corresponding questions. The dataset was prepared from Wikipedia by querying questions beginning with "wh" and ending with a question mark using the Bing search engine. SQuAD stands for Stanford question answering dataset is another popular corpus for machine reading extracted from Wikipedia articles (*Rajpurkar, Jia & Liang, 2018*). The dataset consists of 151,054 questions including 53,775 negative examples from 505 articles. The correct answers in the dataset are contained within the text in the form of sequential tokens. The dataset was prepared by humans through crowdsourcing and contains diverse types of question answers. This study incorporates both SQuAD 1.0 and SQuAD 2.0 for empirical analysis. News-QA is a benchmark text corpus prepared by Microsoft (*Bi et al., 2021*) for reasoning and machine reading comprehension. The dataset consists of 120K question-answer pairs extracted from Cable News Network (CNN) articles. All the questions in the corpus are collected and written by humans in natural language using a 3-stage and siloed mechanism. This dataset is more complex and challenging because most of the questions involve reasoning to answer a question.

## Performance evaluation measures

In this study, we employ two widely used evaluation measures including F1-score, and exact match (EM) to validate the performance of the proposed model. The successful evaluation of the MRC system is tricky due to multiple forms of correct answers. The pipeline of MRC QA consists of retrieval and reader components where the retriever selects a subset of documents from a massive repository and the reader extracts the exact answer. Recall and precision are two frequently used measures for the evaluation of machine learning models. The fraction of relevant answers that are retrieved are termed recall, and the fraction of retrieved answers that are relevant is known as precision. The recall and precision are computed using Eqs. (14) and (15) and respectively:

$$\text{Recall} = \frac{\#(\text{relevant items retrieved})}{\#(\text{total relevant items})} = \frac{Tp}{Tp + Fn} \tag{14}$$

$$\text{Precison} = \frac{\#(\text{relevant items retrieved})}{\#(\text{total retrieved items})} = \frac{tp}{Tp + Fp} \tag{15}$$

where $Tp$ and $Tn$ are denoted as correctly classified relevant and non-relevant text documents. In contrast, $Fp$ and $Fn$ indicate wrongly predicting the relevancy of retrieved documents.

In the evaluation of the reader component, involve F1-score and EM to assess the degree to which the selected answer by the reader module is matched with the correct answer. As the name implies, EM measures the fraction of text document where the model predicted answer is identical to the correct answer based on characters. The F1-score is a more flexible measure than EM as it computed the similarity between the answer predicted by the model and the answer in ground truth based on words. The F1-score, EM, and are computed using Eqs. (16) and (17) respectively:

**Table 2  Hyperparameters for ExyGPT-QA model for machine reading comprehension.**

| Hyper-parameter | Value |
|---|---|
| Learning rate | 3e−05 |
| Train batch size | 8 |
| Eval batch size | 8 |
| Seed | 42 |
| Total train batch size | 16 |
| Total eval batch size | 16 |
| Optimizer | Adam |
| Betas | 0.9,0.999 |
| Epsilon | 1e−08 |
| LR scheduler type | Linear |
| LR scheduler warmup ratio | 0.1 |
| Training steps | 200 |

$$EM = \begin{cases} 1 & \text{if} c_M = a_t \epsilon A_t \\ 0 & \text{otherwise} \end{cases} \qquad (16)$$

$$F1 = \frac{2}{\text{recall}^{-1} + \text{precision}^{-1}} \qquad (17)$$

where, $c_M$ and $a_t$ indicates the characters of the predicted answer and the actual correct answer respectively. Furthermore, recall is a fraction of the shared word to the total words in the correct answer and precision is the proportion of the number of shared words to the total words in the predicted answer.

## RESULTS AND DISCUSSION

This section presents a detailed empirical analysis of the proposed ExtGPT-QA transformers-based model for text comprehension. The results are mainly computed over three MRC benchmarks including SQuAD, WikiQA, and NewsQA using standard evaluation measures. Table 2 summarize the detail of the hyperparameters used for the evaluation of the proposed model.

The learning rate ($lr$) ensures the network's weights are adjusted by following the loss gradient. In *Lapchaicharoenkit & Vateekul (2020)* authors claim that the lower number for the learning rate takes into account every local minima and traversal along the incline slope. Therefore, we set up 0.00003 as the learning rate for ExtGPT-QA to enhance the network's weight modifications. Additionally, the number of samples that will be propagated through the network is determined by the batch size that we employ same 8 as batch size for both training and validation. Correspondingly, the total training batch size and total validation batch sizes are set up as the same 16 for both. A seed in an integer value serves as a helper function to maintain performance stability during dispersed training. Moreover, based on training data, the optimizer repeatedly modifies network weights. To boost up the performance, we espouse Adam as optimizer along with 42 as seed value and betas as 0.9

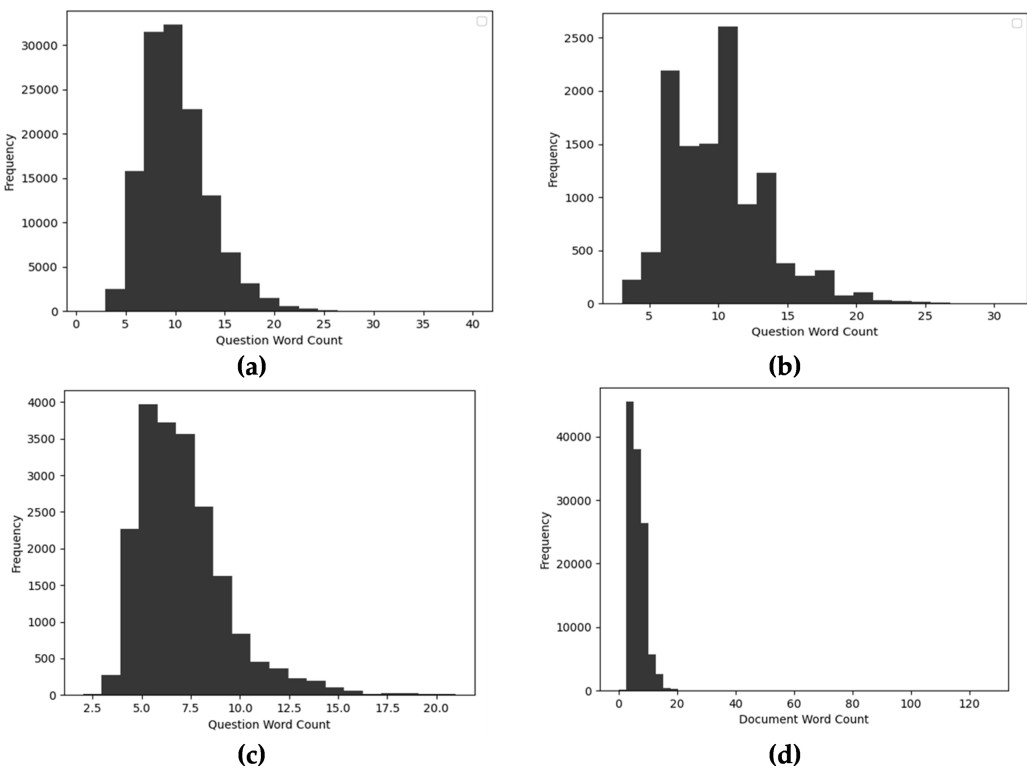

**Figure 9** Distribution of question word count of diverse MRC datasets (A) SQuAD 1.0, (B) SQuAD 2.0, (C) WikiQA, and (D) NewsQA.

and 0.999 respectively. As well, we utilized the linear as *lr* scheduler type and the epsilon value of 0.00000001 to maintain the numerical stability for the Adam optimizer. The *lr* scheduler warmup ratio tunes or increase the learning rate value during the initial stages of training. The basic purpose of warmup is to prevent massive oscillations in the gradients by tuning the learning rate after a certain period gradually to smoothly adjust the optimization landscape. Finally, the ExtGPT-QA model has trained over 200 steps over benchmark MRC datasets.

## Length analysis

Syntax feature engineering involves length-related features such as word count, unique word count, average word length, and n-gram. These features provide significant insights with low computations. Figure 9 presents the word count of each question in diverse datasets. As shown in Figs. 9A and 9B presents the word count of SQuAD 1.0 and SQuAD 2.0 most question is comprising of 5-25 words. However, the maximum word count is about 40 words in SQuAD 1.0 comprehension dataset. Besides that, in the NewsQA dataset, most of the questions have a length of 5-13 words as shown in Fig. 9C. Moreover, Fig. 9D

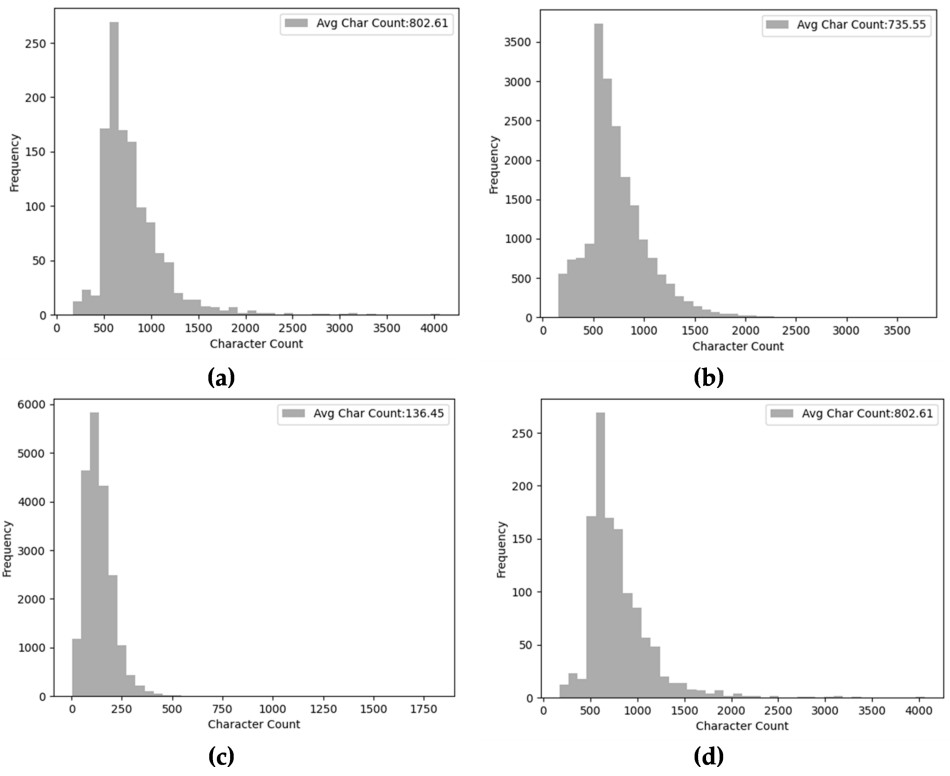

**Figure 10** Distribution of passage unique word count of diverse MRC datasets (A) SQuAD 1.0, (B) SQuAD 2.0, (C) WikiQA, and (D) NewsQA.

depicts the word count of questions in the WikiQA dataset which has a higher word count as compared to other benchmarks.

Moreover, Fig. 10 presents the character count and average character count of each context passage in diverse MRC datasets. As Figs. 10A and 10B depict the charter count of SQuAD corpus shows that most of the passages consist of 100-20000 characters. The average character count is 802 and 735 for SQuAD 1.0 and SQuAD 2.0 respectively. Similarly, the average character count of the NewsQA dataset is 768 and mostly the passage length is ranging between 200 to 1500 characters. In contrast, the WikiQA machine reading comprehension dataset has a less character count of 136 and a passage length ranging up to 600 characters.

## Results of ExtGPT-QA using SQuAD dataset

The experiments were carried out using the proposed ExtGPT-QA model with three different values of learning rate including $3 \times 10^{-5}$, $2 \times 10^{-5}$ and $1 \times 10^{-5}$ over SQuAD 1.0 and SQuAD 2.0 datasets. Results in Table 3 are evident that model attained higher F1 and EM with a lower learning rate. We achieved 93.1% and 90.46% F1 and EM respectively for MRC tasks using SQuAD 1.0 development set. In addition, for the test set, EXTGPT-QA achieved 92.14% and 90.10 F1 and EM respectively using $1 \times 10^{-5}$ learning rate over 200 training steps.

**Table 3 Performance of proposed ExtGPT-QA model over SQuAD 1.0 MRC dataset.**

| Model | Learning rate (LR) | Training steps | SQuAD 1.0 Dev | | SQuAD 1.0 Test | |
|---|---|---|---|---|---|---|
| | | | F1 | EM | F1 | EM |
| E-GPT QA + Affine + Aggregation | 0.00003 | 50 | 76.12 | 71.50 | 72.52 | 70.85 |
| | | 100 | 84.52 | 82.41 | 81.25 | 78.14 |
| | | 150 | 87.41 | 84.63 | 86.45 | 83.69 |
| | | **200** | **88.63** | **86.25** | **87.67** | **85.20** |
| | 0.00002 | 50 | 78.15 | 73.01 | 73.58 | 71.28 |
| | | 100 | 85.57 | 83.46 | 80.55 | 78.14 |
| | | 150 | 87.69 | 85.14 | 88.10 | 86.69 |
| | | **200** | **89.90** | **88.25** | **90.79** | **88.76** |
| | 0.00001 | 50 | 76.10 | 74.59 | 76.50 | 74.28 |
| | | 100 | 89.48 | 85.17 | 89.69 | 86.81 |
| | | 150 | 92.89 | 89.68 | 92.00 | 90.08 |
| | | **200** | **93.1** | **90.46** | **92.14** | **90.10** |

**Notes.**
*Bold values highlight the maximum results achieved after each 200 epochs.

Moreover, Fig. 11 depicts the comparison of the F1 score of the proposed model over the different number of training steps and learning rates. The model achieved about 70% of the F1 score with only 50 training steps and a greater learning rate *i.e.,* $3 \times 10^{-5}$. Similarly, Fig. 12 presents the exact match performance of the proposed model over the development as well as test datasets. Over the 50 training steps model attained 71.51% of an exact match for the MRC task and gradually increase with more training steps and a lower learning rate. However, at 150 training steps, results show a significant increase in EM rates of 90.08%. Likewise, EXTGPT-QA acquired a substantial EM of 86.81% with $1 \times 10^{-5}$ using the test dataset.

Another variant of SQuAD, known as SQuAD 2.0 which contains more complex passages and questions, has been incorporated to assess the performance of the model. Table 4 presents the comparison of ExtGPT-QA over different learning rates and training steps. Our model achieved 92.20% and 90.52% F1 score and EM respectively with 200 training steps and $1 \times 10^{-5}$ learning rate. The performance of the model with a higher learning rate in the initial training steps is lower as compared with the results of the SQuAD 1.0 data set because it is a very difficult data set. The model obtained 63.87% and 60.85% of F1 and EM for the development set and 64.28% and 62.85% of F1 and EM respectively for the test dataset of SQuAD 2.0. However, we found an interesting inverse relationship between learning rate and model performance. As the training steps increase and the learning rate decreases, the performance of the model can be significantly improved.

Figure 13 depicts the comparison of the F1 score of the proposed ExtGPT-QA architecture over different training steps and learning rates over the development and test of the SQuAD 2.0 dataset. The model maintains the inverse relation between learning rate and model performance. However, it can also be broken down that the performance of the model increased from 87.28 to 88.98 due to a 0.00001 decrease in the learning rate at

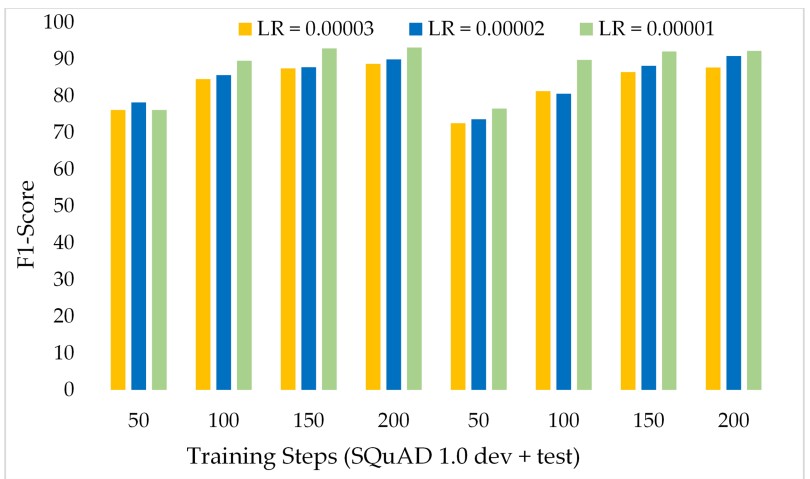

**Figure 11** **Comparison of F1-Score of MRC using proposed ExtGPT-QA over SQuAD 1.0 dataset.**

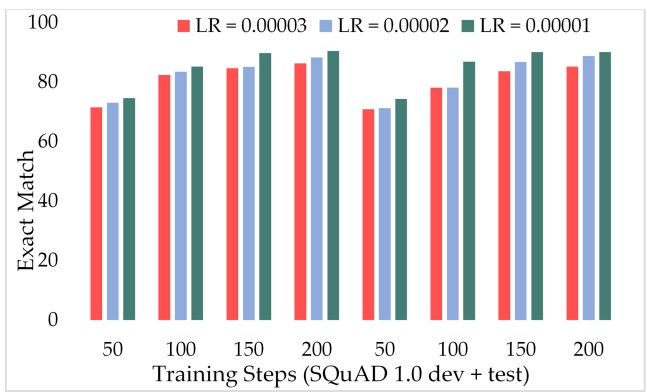

**Figure 12** **Comparison of exact match of MRC using proposed ExtGPT-QA over SQuAD 1.0 dataset.**

200 training steps, an increase of only 1.7%. Correspondingly, Figure 14 illustrates the exact match rates of the proposed model over the SQuAD 2.0 corpus. Although the incorporated dataset contains very complex contexts and unanswerable questions, the model obtained EM values above 60 even at 50 training steps. Further, as the number of training steps increased, the value of EM also increased up to 85.69%. Additionally, when simultaneously reducing the learning rate for the same number of training steps, we got a value of 90.52% EM.

## Results of ExtGPT-QA using Wiki-QA dataset

To assess the performance of our proposed model on lengthy passages, we incorporated the Wiki-QA dataset which is prepared for MRC purposes and contains complex and lengthy passages. Table 5 presents the obtained results from ExtGPT-QA with three different learning rates over the Wiki-QA dataset. Moreover, also provide the comparison of both

**Table 4  Performance of proposed ExtGPT-QA model over SQuAD 2.0 MRC dataset.**

| Model | Learning rate (LR) | Training steps | SQuAD 2.0 Dev | | SQuAD 2.0 test | |
|---|---|---|---|---|---|---|
| | | | F1 | EM | F1 | EM |
| E-GPT QA + Affine + Aggregation | 0.00003 | 50 | 63.87 | 60.85 | 64.28 | 62.85 |
| | | 100 | 78.58 | 76.69 | 79.43 | 77.28 |
| | | 150 | 88.10 | 84.20 | 86.43 | 82.82 |
| | | **200** | **87.28** | **85.69** | **88.59** | **84.36** |
| | 0.00002 | 50 | 64.82 | 62.18 | 68.10 | 66.79 |
| | | 100 | 78.89 | 77.28 | 80.10 | 78.69 |
| | | 150 | 88.10 | 86.76 | 86.92 | 86.34 |
| | | **200** | **88.98** | **87.28** | **89.48** | **87.69** |
| | 0.00001 | 50 | 70.28 | 68.95 | 71.36 | 69.37 |
| | | 100 | 79.56 | 78.39 | 81.25 | 80.28 |
| | | 150 | 90.12 | 89.20 | 88.40 | 87.58 |
| | | **200** | **92.20** | **90.52** | **91.29** | **90.00** |

**Notes.**
*Bold values highlight the maximum results achieved after each 200 epochs.

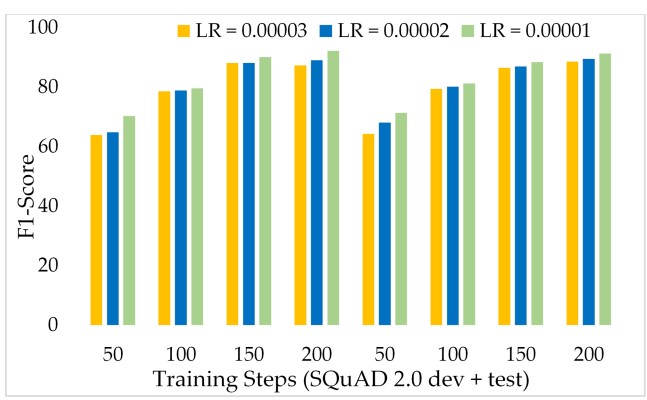

**Figure 13  Comparison of F1 Score of MRC using proposed ExtGPT-QA over SQuAD 2.0 dataset.**

development and test sets with gradually increasing training steps. Results are evident that our model performed well on long passages and reached up to 93.25% and 91.05% F1 score and EM respectively. Even, around the 50 training steps our model attained 71.50% and 69.84% F1 and EM correspondingly on the development set and 70.36% and 68.25% F1 and EM rate respectively on the test dataset. Moreover, with a learning rate of $2 \times 10^{-5}$ and 200 training steps the ExtGPT-QA achieves about 90.12% and 90.83% of the F1 score over the development and test dataset accordingly.

Figure 15 presents the comparison of the F1 score of ExtGPT-QA using the development and test sets of the Wiki-QA dataset over different learning rates and training steps. We found an inverse relationship between the F1 score and learning rate, as the lower learning rate model attained a better F1 score. However, the training steps are directly proportional

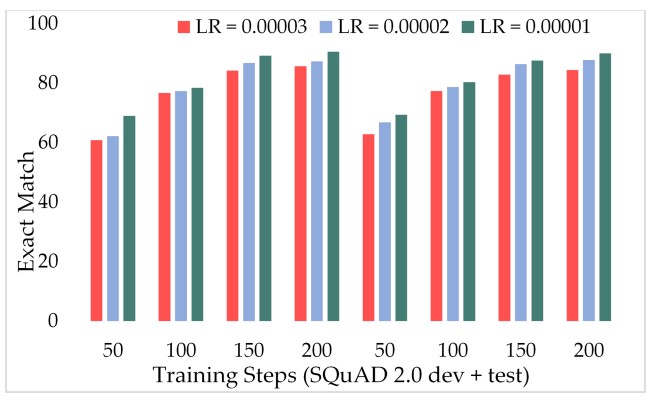

**Figure 14** Comparison of exact match of MRC using proposed ExtGPT-QA over SQuAD 2.0 dataset.

**Table 5** Performance of proposed ExtGPT-QA model over Wiki-QA MRC dataset.

| Model | Learning rate (LR) | Training steps | WikiQA Dev | | WikiQA test | |
|---|---|---|---|---|---|---|
| | | | F1 | EM | F1 | EM |
| | | 50 | 71.56 | 69.84 | 70.36 | 68.25 |
| | 0.00003 | 100 | 76.80 | 75.39 | 79.10 | 77.63 |
| | | 150 | 87.28 | 84.80 | 87.43 | 85.96 |
| | | **200** | **89.70** | **87.69** | **89.00** | **86.69** |
| | | 50 | 72.93 | 70.64 | 71.39 | 69.40 |
| E-GPT QA + Affine + Aggregation | 0.00002 | 100 | 77.69 | 76.98 | 80.90 | 79.01 |
| | | 150 | 89.10 | 86.76 | 86.92 | 86.34 |
| | | **200** | **90.12** | **88.69** | **90.83** | **88.69** |
| | | 50 | 72.52 | 70.86 | 73.69 | 71.25 |
| | 0.00001 | 100 | 81.27 | 79.27 | 82.18 | 79.98 |
| | | 150 | 91.05 | 89.39 | 90.86 | 98.58 |
| | | **200** | **93.25** | **91.05** | **92.13** | **90.48** |

**Notes.**
*Bold values highlight the maximum results achieved after each 200 epochs.

to the F1 score, as we higher training steps model performed well. Despite the length of context passages, ExtGPT-QA obtained 93.25 and 92.13 F1-score for the development and test datasets respectively. Moreover, Figure 16 illustrates the exact match of the proposed model using the Wiki-QA dataset over different learning rates and training steps. According to the results of the proposed model, in the initial steps around 50, the exact match remains almost constant for all three learning rates, however, after 70 training steps the value of EM increases gradually with a lower learning rate. In the result, it can be seen that the model obtained the most exact match on the test data set with 150 training steps and $1 \times 10^{-5}$ as the learning rate.

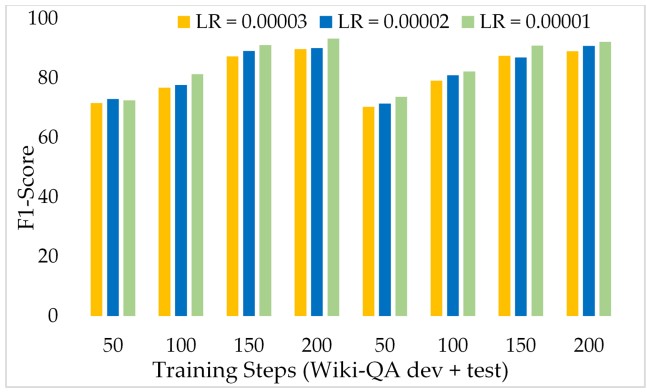

**Figure 15** Comparison of F1 Score of MRC using proposed ExtGPT-QA over Wiki-QA dataset.

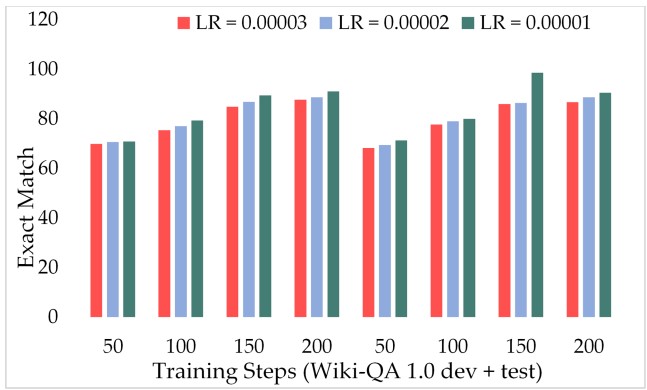

**Figure 16** Comparison of exact match of MRC using proposed ExtGPT-QA over Wiki-QA dataset.

## Results of ExtGPT-QA using News-QA dataset

The News-QA dataset contains paragraphs and questions that require reasoning and human-level reading comprehension to answer. Additionally, it contains some questions that have no exact answers in context passages and require reasoning skills to answer the question. We used this dataset to assess the effectiveness of the proposed model and the validation of results for the MRC task. Table 6 presents the performance of our ExtGPT-QA model over News-QA development and test dataset using different learning rates and training steps. We obtained 91.20% and 90.28% F1-score and EM respectively for the development set and 91.50% and 89.58% F1-score and EM correspondingly for the test dataset. We observe two interesting relations, firstly direct relation between the training steps model's performance and secondly inverse relation between learning rate and performance. According to the results, the model attained 89.69% and 88.97% F1 and EM respectively with 200 training steps and $3 \times 10^{-5}$ as the learning rate. Likewise, for lower learning $2 \times 10^{-5}$ with the same training steps, we obtained 90.58% and 89.79% F1-score and EM respectively.

**Table 6  Performance of proposed ExtGPT-QA model over news-QA MRC dataset.**

| Model | Learning rate (LR) | Training steps | NewsQA Dev | | NewsQA test | |
|---|---|---|---|---|---|---|
| | | | F1 | EM | F1 | EM |
| E-GPT QA + Affine + Aggregation | 0.00003 | 50 | 68.79 | 66.79 | 69.10 | 67.79 |
| | | 100 | 75.18 | 73.28 | 76.45 | 74.59 |
| | | 150 | 86.89 | 85.07 | 88.42 | 87.17 |
| | | **200** | **89.46** | **88.97** | **89.69** | **89.25** |
| | 0.00002 | 50 | 69.86 | 67.28 | 70.81 | 67.57 |
| | | 100 | 76.17 | 74.83 | 78.18 | 77.59 |
| | | 150 | 87.59 | 86.49 | 88.93 | 87.34 |
| | | **200** | **90.58** | **89.79** | **90.00** | **89.20** |
| | 0.00001 | 50 | 71.25 | 69.47 | 72.38 | 69.79 |
| | | 100 | 81.82 | 80.10 | 84.28 | 82.19 |
| | | 150 | 88.40 | 86.49 | 89.09 | 87.39 |
| | | **200** | **91.20** | **90.28** | **91.50** | **89.58** |

**Notes.**
*Bold values highlight the maximum results achieved after each 200 epochs.

Figure 17 presents the comparison of the F1 Score for the MRC task using the proposed ExtGPT-QA over News-QA dataset. Results are evident that the proposed model maintained a constant and direct relationship between the F1 score and model performance. At the initial training steps, we observed about 69% F1 with a learning rate of $3 \times 10^{-5}$. Furthermore, the model reached to 91.20% F1 score with 200 training steps and $1 \times 10^{-5}$ such a complex dataset. Similarly, Figure 18 shows the comparison of an exact match for comprehension tasks using the proposed EXTGPT-QA over the News-QA dataset. In the first 100 training steps, the exact match of the model increased significantly and observed an inversely proportional relationship between learning rate and performance. However, the EM of all three learning rates remained the same for NEWS-QA in contrast to all other datasets with training steps above 100. Despite the complexity of the dataset, our model obtained 90.28% and 89.58 for the development and test dataset respectively.

## Comparison of the ExtGPT-QA with state-of-art MRC models

We compare our proposed model against recent MRC approaches on three benchmark datasets to validate the architecture and assess the effectiveness of the comprehension task. According to Table 7, BERT with NeurQuRI (*Back et al., 2020*) obtained about 73% and 70% F1 score and EM, respectively for SQuAD and News-QA datasets. Likewise, BERT with bidirectional LSTM (*Ji et al., 2022*) reached up to 83% and 80% of the F1-score and EM rate correspondingly. Moreover, another variation of BERT with ProphetNet (*Aithal, Rao & Singh, 2021*) achieved 86.79% and 84.40% F1 and EM, respectively, over the SQuAD dataset. Similarly, BERT with CLSTM and MTL (*He et al., 2022*) reached up to 88% and 84% of the F1 score over two MRC datasets, including SQuAD and Wiki-QA. The BERT model along with discourse-aware self-attention (*Galitsky, Ilvovsky & Goncharova,*

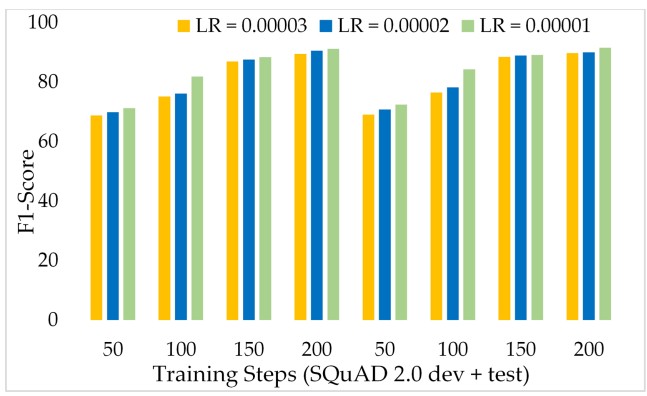

**Figure 17  Comparison of F1 score of MRC using proposed ExtGPT-QA over News-QA dataset.**

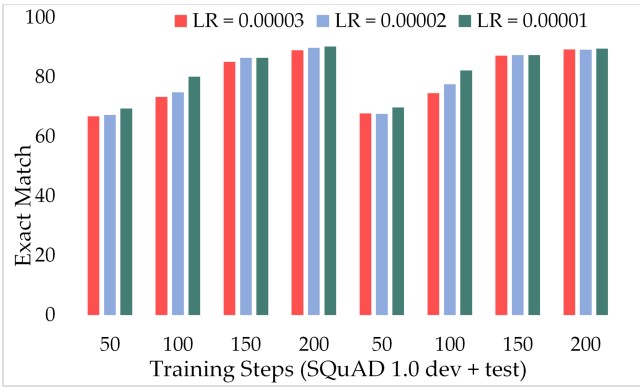

**Figure 18  Comparison of exact match of MRC using proposed ExtGPT-QA over News-QA dataset.**

*2021*) performed well on the News-QA dataset as compared to the SQuAD model and obtained an 82.85% exact match. Furthermore, another transformers-based model TANDA along with RoBERTa (*Garg, Vu & Moschitti, 2019*) achieved more than 90% EM and F score, far surpassing human comprehension. The SpanBERT (*Joshi et al., 2020*) model achieved 92.08% and 90.20% F1 and EM, respectively for News-QA and SQuAD datasets. Likewise, RLAS-BIABC (*Gharagozlou et al., 2022*) applied over two complicated datasets including SQuAQ 2.0 and Wiki-QA and obtained 92.40% and 90.80% F1 score and exact match, respectively. However, our proposed model outperformed all BERT variations and transformers-based state-of art MRC techniques. The E-GPT achieved up to 93.25% and 90.52% F1-score by mitigating the textual ambiguities and semantic vagueness issues with the machine reading comprehension task.

The results of the proposed model are in complete alignment with the contributions and objectives of this research study. The first objective of the study was to design an extended architecture based on the GPT base model that would be able to answer MRC-based questions. All the research experiments are carried out using a novel proposed ExtGPT-QA model using three MRC datasets. Similarly, the second objective was to address the issues

**Table 7** Comparison of the proposed ExtGPT-QA model with existing MRC methods over SQuAD, Wiki-QA, and News-QA datasets.

| Model/Method | SQuAD 1.0 | | SQuAD 2.0 | | Wiki-QA | | NewsQA test | |
|---|---|---|---|---|---|---|---|---|
| | F1 | EM | F1 | EM | F1 | EM | F1 | EM |
| DocQA (ELMo) + NeurQuRI (*Back et al., 2020*) | 73.80 | 70.50 | 72.8 | 69.7 | – | – | 73.28 | 70.25 |
| BERT + NeurQuRI (*Back et al., 2020*) | 82.90 | 80.00 | 83.10 | 80.0 | – | – | 83.62 | 79.28 |
| BERT + bidirectional-LSTM (*Ji et al., 2022*) | – | – | 82.39 | 79.50 | 82.53 | 79.68 | 81.60 | 79.58 |
| BERT + ProphetNet (*Aithal, Rao & Singh, 2021*) | 86.79 | 84.40 | 85.28 | 83.00 | – | – | – | – |
| BERT + CLSTM + MTL (*He et al., 2022*) | 88.20 | 84.90 | – | – | 84.8 | 82.93 | – | – |
| BERT + Discourse aware Self Attention (*Galitsky, Ilvovsky & Goncharova, 2021*) | 82.10 | 80.29 | 80.49 | 79.53 | – | – | 84.39 | 82.85 |
| TANDA-RoBERTa (*Garg, Vu & Moschitti, 2019*) | 92.30 | 90.10 | – | – | 91.20 | 90.63 | 89.40 | 87.63 |
| SpanBERT (*Joshi et al., 2020*) | 92.08 | 90.52 | 91.52 | 90.20 | – | – | 90.63 | 88.91 |
| RLAS-BIABC (*Gharagozlou et al., 2022*) | – | – | 91.93 | 89.40 | 92.40 | 90.80 | – | – |
| Proposed E-GPT QA | **93.1** | **90.46** | **92.20** | **90.52** | **93.25** | **91.05** | **91.20** | **90.28** |

**Notes.**
*Bold values highlight the maximum results achieved after each 200 epochs.

of textual ambiguity and semantic vagueness in the MRC task. For this purpose, we selected datasets that were specifically prepared for MRC tasks, which had both linguistic issues. Therefore, the results of the proposed model serve as evidence that we have effectively solved these problems to a significant extent. Table 8 presents the results of the proposed model on a passage selected from SQuAD 2.0 MRC dataset. This passage contains textual ambiguities in the phrase *"some plant-based diets may not provide sufficient amounts of certain nutrients"*. It is unclear whether the diets themselves are deficient in these nutrients or whether people who follow the diets may not be consuming enough of these nutrients. Moreover, the passage also contains semantic vagueness in the phrase *"carefully plan their meals to ensure they are meeting their nutrient needs"*. It is not clear what specific steps someone would need to take to ensure they are meeting their nutrient needs, and this could vary depending on individual factors like age, gender, and activity level. The results show that ExtGPT generates the correct answers to five questions with an average 91.4% exact match and 94.8% F1-score. Additionally, our objectives included the use of three benchmark MRC datasets, namely SQuAD, Wiki-QA, and News-QA, for empirical analysis. The results of this analysis are discussed in the above sections and also presented in Tables 3–6. The final objective was to compare the proposed model with state-of-the-art MRC models, therefore, Table 7 presents the comprehensive comparison of ExtGPT-QA against other models. However, the results are evidence that the proposed model outperformed other MRC models.

# CONCLUSIONS

The proposed model, ExtGPT-QA, uses encoder decoder transformers-based architecture for the selection of answers to posed questions from a specific context. The first objective was to extend the GPT model, therefore, we introduced two additional layers *i.e.,* affine and aggregation layers with GPT architecture to increase the effectiveness of MRC task.

**Table 8    Results of ExtGPT-QA model on example passage from the SQuAD 2.0 dataset.**

Passage: *"Plant-based diets are often associated with a lower risk of chronic diseases such as heart disease and cancer. However, research has shown that some plant-based diets may not provide sufficient amounts of certain nutrients, such as vitamin B12, calcium, and iron. Therefore, it is important for individuals following a plant-based diet to carefully plan their meals to ensure they are meeting their nutrient needs".*

| Q. No. | Question | Actual answer | Predicted by Ext-GPT model | Results | |
| --- | --- | --- | --- | --- | --- |
| | | | | E.M | F1 |
| Q1. | What are some chronic diseases that plant-based diets may help reduce the risk of? | Plant-based diets are often associated with a lower risk of chronic diseases such as heart disease and cancer. | Plant-based diets are associated with a lower risk of heart disease and cancer. | 86.2 | 95.40 |
| Q2. | What are some nutrients that may be deficient in certain plant-based diets? | Some plant-based diets may not provide sufficient amounts of certain nutrients, such as vitamin B12, calcium, and iron. | Certain plant-based diets may not provide enough vitamin B12, calcium, and iron. | 89.23 | 94.16 |
| Q3. | Why is it important for individuals following a plant-based diet to carefully plan their meals? | It is important for individuals following a plant-based diet to carefully plan their meals to ensure they are meeting their nutrient needs. | Individuals following a plant-based diet should plan their meals carefully to ensure they are getting the nutrients they need. | 91.20 | 96.37 |
| Q4. | What are some nutrients that people following a plant-based diet may need to pay extra attention to? | People following a plant-based diet may need to pay extra attention to getting enough vitamin B12, calcium, and iron. | Vitamin B12, calcium, and iron are nutrients that people following a plant-based diet may need to pay extra attention to. | 93.10 | 92.60 |
| Q5. | What is unclear about the phrase "some plant-based diets may not provide sufficient amounts of certain nutrients"? | It's unclear whether the diets themselves are deficient in these nutrients or whether people who follow the diets may not be consuming enough of these nutrients. | It's unclear whether the diets themselves are deficient in these nutrients or whether people who follow the diets may not be getting enough of these nutrients. | 98.10 | 97.20 |

To comply the second objective of addressing the issues of textual ambiguity and semantic vagueness, the model combined syntax-based and semantic-based features for a rich understanding of natural language text along with a multi-head attention mechanism. To achive the third objective of evaluation of the proposed model, the results are computed on three benchmark MRC datasets to examine the performance of the proposed ExtGPT-QA against the state of art models. The detailed empirical analysis reveals that our model outperformed recent deep learning and transformer-based models for MRC question answering. In a nutshell, the ExtGPT-QA model over the 200 training steps and $1 \times 10^{-5}$ learning rate higher exact March and F1-score rates as compared with higher learning rates. Results emerged an interesting inverse relationship between model performance and learning rate. For instance, with $3 \times 10^{-5}$ and $1 \times 10^{-5}$ model attained 89.70% and 93.25% F1 scores, respectively over the same training steps over the WikiQA dataset. On News-QA, which is considered a complex and challenging MRC dataset, our model showed excellent performance, which can be the basis of our model's effectiveness. The last objective of comparative analysis was achived as the results showed that the proposed ExtGPT-QA model effectively tackled textual ambiguity and semantic vagueness challenges

and achieved the stated goals. Practically, the proposed E-GPT model can be applied for the search engine to respond to users' queries with exact answers in natural language instead of a ranked list of web pages. The future works of this study include resolving the natural language ambiguities with logical reasoning and global knowledge to enhance the effectiveness MRC systems. In addition, we plan to involve knowledge graph-based methods with the ExtGPT-QA model to extend its capabilities for multi-passage MRC systems.

### Funding
This work was supported by the Deanship of Scientific Research, Vice Presidency for Graduate Studies and Scientific Research, King Faisal University, Saudi Arabia [Grant No. 2051]. Both equally contributed in overall research study from method formulation to study design, data collection, empirical analysis, reviews and analysis.

### Grant Disclosures
The following grant information was disclosed by the authors:
Deanship of Scientific Research, Vice Presidency for Graduate Studies and Scientific Research, King Faisal University, Saudi Arabia: Grant No. 2051.

### Competing Interests
The authors declare there are no competing interests.

### Author Contributions
- Muzamil Ahmed conceived and designed the experiments, performed the experiments, performed the computation work, prepared figures and/or tables, and approved the final draft.
- Hikmat Khan conceived and designed the experiments, performed the experiments, performed the computation work, prepared figures and/or tables, authored or reviewed drafts of the article, and approved the final draft.
- Tassawar Iqbal conceived and designed the experiments, performed the experiments, analyzed the data, performed the computation work, prepared figures and/or tables, and approved the final draft.
- Fawaz Khaled Alarfaj analyzed the data, authored or reviewed drafts of the article, and approved the final draft.
- Abdullah Alomair analyzed the data, authored or reviewed drafts of the article, and approved the final draft.
- Naif Almusallam analyzed the data, authored or reviewed drafts of the article, and approved the final draft.

### Data Availability
  The three MRC datasets are available in the Supplemental Files.

Dataset 1: Stanford Question Answering Dataset (SQuAD), Stanford University NLP Group, DOI: 10.5555/3275533.3275543, https://rajpurkar.github.io/SQuAD-explorer/.

Dataset 2: WikiQA, Microsoft Research, DOI: 10.1145/2641048.2641269, https://www.microsoft.com/en-us/download/details.aspx?id=52419.

Dataset 3: NewsQA, Microsoft, University of Washington, the Allen Institute for Artificial Intelligence (AI2), and the University of Michigan, DOI: 10.1007/s10579-019-09437-2, https://www.microsoft.com/en-us/research/project/newsqa.

Code implementation Python files are available in the Supplemental Files.

## Supplemental Information

Supplemental information for this article can be found online at http://dx.doi.org/10.7717/peerj-cs.1422#supplemental-information.

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
