# Peer review of "On solving textual ambiguities and semantic vagueness in MRC based question answering using generative pre-trained transformers"

_PeerJ Computer Science, doi:10.7717/peerj-cs.1422_

## Round 0.1 · original submission · Minor Revisions

Dear Authors,

Thank you for submitting your manuscript to PeerJ Computer Science.

We have completed the evaluation of your manuscript. The reviewers recommend reconsideration of your manuscript following minor revisions. I invite you to resubmit your manuscript after addressing the comments made by the reviewers. In particular:
1. Perform a general review of the writing (several changes are indicated by reviewer 1 and a few additional changes are indicated by reviewer 2).
2. Explicitly link results to your hypotheses and research questions.
3. Summarize goals and contributions in the conclusion section.
I hope you can complete the recommendation changes in a revision of your article.

Best,
Ana Maguitman

·

Basic reporting

General comments on the manuscript:

The present research addresses two key challenges in Machine Reading Comprehension (MRC) systems based on Natural Language Processing (NLP): textual ambiguities and semantic vagueness. These challenges hinder the understanding of lengthy passages and the generation of answers in abstractive MRC systems. To tackle these issues, a novel Extended Generative Pretrained Transformers-based Question Answering (ExtGPT-QA) model is proposed, which aims to generate precise and relevant answers to questions related to a given context.

The ExtGPT-QA model features an architecture that includes two modified forms of encoder and decoder compared to the original GPT. The encoder uses a positional encoder to assign a unique representation to each word in the sentence, which helps address textual ambiguities. On the other hand, the decoder employs a multi-head attention mechanism along with an affine and aggregation layer to mitigate semantic vagueness in MRC systems. Furthermore, syntax and semantic feature engineering techniques are applied to enhance the effectiveness of the proposed ExtGPT-QA model.

The main contributions of this work include the proposal and development of the ExtGPT-QA model and the application of feature engineering techniques to optimize its effectiveness. To validate the effectiveness of the proposed model, a comprehensive empirical analysis is carried out using three benchmark datasets: SQuAD, Wiki-QA, and News-QA. The results obtained by the ExtGPT-QA model outperform state-of-the-art MRC techniques, achieving a 93.25% F1 score and a 90.52% exact match, demonstrating its capability to address textual ambiguities and semantic vagueness in MRC systems.


Specified comments and evaluation:

1. Clear and unambiguous, professional English used throughout: In general, the use of English is clear and professional in most of the text. However, there are some sentences that could benefit from revision and correction to improve clarity and flow. It would be useful to go through the entire text to ensure that there are no grammatical errors, typos or confusing phrases.

2. Self-contained with relevant results to hypotheses: The text appears to be self-contained and presents relevant results related to the hypotheses stated. The ExtGPT-QA model is compared with other state-of-the-art models, and results on various CRM datasets are presented to demonstrate its effectiveness. Furthermore, practical implications and possible future improvements are discussed. However, it would be useful to ensure that the results are clearly linked to the hypotheses or research questions posed and that all proposed goals and contributions are addressed.

The following points are considered appropriate and are well-drafted in the manuscript:

3. Literature references, sufficient field background/context provided.
4. Professional article structure, figures, tables. Raw data shared.
5. Formal results should include clear definitions of all terms and theorems, and detailed proofs.

Experimental design

The originality of the research is appropriate to the objectives of this journal. In addition, the research questions are well defined. The research is sufficiently rigorous.

Validity of the findings

I am grateful to the authors for sharing the code and datasets ensuring the reproducibility of the research and the experiments performed. On the other hand, I suggest an improvement related to the conclusions of the manuscript:

1. To ensure that the conclusions are fully aligned with the stated goals and contributions, it would be useful to compare them directly with the goals and contributions section at the beginning of the paper. Make sure that all stated goals and proposed contributions are addressed in the conclusions and that it is explained how they were achieved or to what extent they were met. It is also important to highlight the unique and novel aspects of the ExtGPT-QA model in the conclusions, in relation to the stated goals and contributions, so that readers can clearly understand the value and importance of this work in the context of CRM research.

Additional comments

In addition, some changes related to the English language of the manuscript are suggested. Only two sections have been evaluated in depth, so authors are encouraged to send the manuscript to an English reviewer to proofread the entire manuscript. These are some of the proposed changes, but not the only ones:

1. Change "The MRC meditates the machine’s ability" to "MRC evaluates the machine's ability".
2. Change "The MRC task has been found to be challenging" to "The MRC task is challenging".
3. In the sentence "As stated in [6], machine-generated responses meet two criteria", change "meet" to "must meet".
4. Change "Prior MRC studies reveal that" to "Previous MRC studies have shown that".
5. Replace "such models do not have generalization capability" with "such models lack generalization capabilities".
6. Change "After the advent of deep learning" to "With the advent of deep learning".
7. Change "In contrast to sequential models, bidirectional models" to "Unlike sequential models, bidirectional models".
8. Replace "Although, before the advent of MRC systems" with "Before the advent of MRC systems, however,".
9. Change "Moreover, gives search engines the ability" to "Moreover, MRC systems give search engines the ability".
10. Change "Over time, MRC systems have improved considerably due to deep learning and pretrained language models but still suffer from linguistic issues" to "Despite significant improvements in MRC systems over time, due to deep learning and pretrained language models, they still suffer from linguistic issues".
11. Change "truing test" to "Turing test" and "turing" should always be capitalised.
Replace "the human can communicate" with "a human can communicate".
12. In the sentence "In the turing test also called the imitation game", capitalize "Turing" and add a comma after "test".
13. Change "question using similarity matching based techniques" to "questions using similarity matching-based techniques".
14. Replace "vectorization technique in which applied" with "vectorization technique that applied".
15. Change "They achieved 80% accuracy on BNPParibas corpus" to "They achieved an 80% accuracy on the BNPParibas corpus".
16. In the sentence "Moreover, the TF-IDF and cosine similarity lack to consider semantic as they focus on frequency-based similarity", replace "lack to consider" with "do not consider".
17. Change "The question answering system lacks common-sense ability" to "The question-answering system lacks the common-sense ability".
18. Replace "compute the similarity between posed question" with "compute the similarity between the posed question".
19. Change "They select 1000 unanswerable and irrelevant question form" to "They selected 1,000 unanswerable and irrelevant questions from".
20. Replace "word segmentation and stop-word-removal" with "word segmentation and stop-word removal".
21. In the sentence "The CNN architecture consists of four convolution layers", add "the" before "four".
22. Replace "another question answering system" with "Another question-answering system".
23. Change "capture sentence past dependencies" to "capture sentence's past dependencies".
24. Replace "used bi-LSTM and BERT" with "used Bi-LSTM and BERT".
Change "The question answer process involves" to "The question-answering process involves".
25. Replace "attained 64.48% and 49.39% MAP score" with "attained MAP scores of 64.48% and 49.39%".
26. Change "The use of attention mechanism improves the accuracy" to "The use of an attention mechanism improves the accuracy".
27. Change "the ensembled transformer-based model" to "the ensemble transformer-based model".
28. Replace "semantic and word position in questions and answers" with "semantics and word positions in questions and answers".
29. Change "Another, BLSTM, SRLP question answering system" to "Another question-answering system, BLSTM-SRLP".
30. Change "Inferential MRC intends to answer posed generic question" to "Inferential MRC aims to answer posed generic questions".
31. Replace "They used RACE, MC-Test and Multi-RC corpus" with "They used the RACE, MC-Test, and Multi-RC corpora".
32. Change "The multi-passage MRC retrieves the answer" to "Multi-passage MRC retrieves the answers".
33. Change "Sentence segmentation from multiple documents" to "Sentence segmentation across multiple documents".
34. Change "They used Du-Reader version 2 corpus" to "They used the DuReader version 2 corpus".
35. Replace "The answer availability prediction in MRC is essential to access" with "Predicting answer availability in MRC is essential to assess".

·

Basic reporting

The article is written clearly. It is well structured, including Introduction, Related Works, Research Methodology, Results and Discussion, and Conclusion sections. Figures are of high quality. The authors should check punctuation, as many questions miss commas. For example, “Moreover, another variation of BERT with ProphetNet [25] achieved 86.79% and 84.40% F1 and EM, respectively, over the SQuAD dataset. Similarly, BERT with CLSTM and MTL [6] reached up to 88% and 84% of the F1 score over two MRC datasets, including SQuAD and Wiki-QA.” No commas before words “respectively” and “including.”

Experimental design

The paper uses an advanced ExtGPT-QA transformers model for the question-answering problem. The architecture and functionalities of this model are described well.

Validity of the findings

The paper showed a large number of experimental results with supportive graphics and plots estimating the quality of designed GPT models. Several datasets were used to check the model’s performance.

Additional comments

No comment

---

## Round 0.2 · accepted · Accept

Thank you for your contribution to PeerJ Computer Science and for systematically addressing all the reviewers' suggestions. The reviewers are satisfied with the revised version of your manuscript.

·

Basic reporting

I would like to express my gratitude to the authors for considering all suggestions provided.

In my opinion, the manuscript has undergone significant enhancements as a result of these revisions.

Experimental design

No comment.

Validity of the findings

No comment.

·

Basic reporting

Thank you for providing the review of your paper

Experimental design

No comment

Validity of the findings

No comment

Additional comments

No comment